# Prolonged Door-to-Balloon Time Leads to Endothelial Glycocalyx Damage and Endothelial Dysfunction in Patients with ST-Elevation Myocardial Infarction

**DOI:** 10.3390/biomedicines11112924

**Published:** 2023-10-29

**Authors:** Carl Vahldieck, Benedikt Fels, Samuel Löning, Laura Nickel, Joachim Weil, Kristina Kusche-Vihrog

**Affiliations:** 1Department of Anesthesiology and Intensive Care Medicine, University Medical Centre Schleswig-Holstein Campus Luebeck, 23538 Luebeck, Germany; 2Institute of Physiology, University of Luebeck, 23562 Luebeck, Germany; benedikt.fels@uni-luebeck.de (B.F.); kristina.kuschevihrog@uni-luebeck.de (K.K.-V.); 3DZHK (German Research Centre for Cardiovascular Research), Partner Site Hamburg/Luebeck/Kiel, 23562 Luebeck, Germany; 4Medizinische Klinik II, Sana Kliniken Luebeck, 23560 Luebeck, Germanyjoachim.weil@sana.de (J.W.)

**Keywords:** endothelial glycocalyx, endothelial dysfunction, ST-elevation myocardial infarction (STEMI), door-to-balloon time, nitric oxide

## Abstract

Damage to the endothelial glycocalyx (eGC) has been reported during acute ischemic events like ST-elevation myocardial infarction (STEMI). In STEMI, a door-to-balloon time (D2B) of <60 min was shown to reduce mortality and nonfatal complications. Here, we hypothesize that eGC condition is associated with D2B duration and endothelial function during STEMI. One hundred and twenty-six individuals were analyzed in this study (STEMI patients vs. age-/sex-matched healthy volunteers). After stimulating endothelial cells with patient/control sera, the eGC’s nanomechanical properties (i.e., height/stiffness) were analyzed using the atomic force microscopy-based nanoindentation technique. eGC components were determined via ELISA, and measurements of nitric oxide levels (NO) were based on chemiluminescence. eGC height/stiffness (both *p* < 0.001), as well as NO concentration (*p* < 0.001), were reduced during STEMI. Notably, the D2B had a strong impact on the endothelial condition: a D2B > 60 min led to significantly higher serum concentrations of eGC components (syndecan-1: *p* < 0.001/heparan sulfate: *p* < 0.001/hyaluronic acid: *p* < 0.0001). A D2B > 60 min led to the pronounced loss of eGC height/stiffness (both, *p* < 0.001) with reduced NO concentrations (*p* < 0.01), activated the complement system (*p* < 0.001), and prolonged the hospital stay (*p* < 0.01). An increased D2B led to severe eGC shedding, with endothelial dysfunction in a temporal context. eGC components and pro-inflammatory mediators correlated with a prolonged D2B, indicating a time-dependent immune reaction during STEMI, with a decreased NO concentration. Thus, D2B is a crucial factor for eGC damage during STEMI. Clinical evaluation of the eGC condition might serve as an important predictor for the endothelial function of STEMI patients in the future.

## 1. Introduction

The endothelial glycocalyx (eGC) and the cellular cortex (CTX), positioned on the luminal side of endothelial cells (ECs), provide a vasoprotective nanobarrier and responsive hub. The eGC and CTX are highly dynamic and can adapt their mechanical properties (stiffness or height) to environmental changes in the bloodstream. This responsiveness of ECs is imperative for proper functioning of the endothelium; damage to the mechanical properties of either the eGC or CTX might induce endothelial dysfunction [1]. A possible cause of such alterations of the endothelial surface is ischemia-reperfusion (I/R) injury (IRI), a protracted but reversible interruption in blood supply and tissue oxygenation, leading to organ damage [2,3]. IRI causes rapid dismantling of the microvascular eGC in all tissues, and its degradation might be the earliest form of structural damage in I/R [2].

Isolated eGC damage in the context of cardiac IRI has been demonstrated in the setting of acute myocardial infarction, both in experimental and clinical settings [4,5], but unfortunately, the functionality of the endothelium itself has not been taken into account in these studies. In ST-elevation myocardial infarction (STEMI), rapid revascularization was shown to reduce in-hospital and long-term mortality, as well as decrease the number of nonfatal complications [6,7]. The time interval between a patient entering the medical system and revascularization to open the occluded, culprit vessel is termed “Door-to-Balloon”-time (D2B). The preferred mode of revascularization is a primary percutaneous coronary intervention (PCI) and, as recommended by the European Society of Cardiology (ESC), the gold standard time from hospital entry to PCI (D2B) is ≤60 min for PCI-capable hospitals, to prevent ischemic cardiogenic shock from progressing further [8,9]. By implementing direct access to catheterization laboratories and bypassing the emergency department, the time from first medical contact to PCI has been reduced over the past few years, and there is a general consensus that a shorter D2B is associated with a better prognosis [8,9].

Successful restoration of the coronary blood flow can be achieved in over 95% of PCI procedures but, despite angiographically complete coronary artery patency, in about half of the patients, perfusion to the distal coronary microvasculature is not fully restored [10]. The exact pathophysiological mechanism of this post-ischemic coronary microvascular dysfunction is still debated [10] and, as far as we know, up to this point there have been no published findings that consider the eGC condition in this important context.

One frequently named biomarker, acting as surrogate parameter for the eGC’s condition, is syndecan-1. Syndecan-1 (Syn-1; CD138), a transmembrane proteoglycan that builds the structural backbone of the eGC, is damaged during IRI [2]. It acts as a core unit for other eGC components, such as heparan sulfate and hyaluronic acid, and transmits extracellular signals to the intracellular environment of endothelial cells through its transmembrane domain, which is directly associated with the actin cytoskeleton of the endothelial cell cortex [11,12]. Although syndecan-1 has often been used as a biomarker of eGC damage after revascularization, as it is independently associated with 6-month mortality after STEMI [13], to the best of our knowledge, there are no data on a temporal relationship between syndecan-1 levels and revascularization time in the event of STEMI.

Up to now, the relationship between eGC condition, D2B duration, and endothelial dysfunction has been unknown. Although some biomarkers for endothelial damage are now established in the context of other diseases (such as sepsis [14]), they have so far remained almost unnoticed in the context of acute cardiac ischemia and have not yet found their way into everyday clinical practice. The purpose of this study therefore was to investigate whether a prolonged D2B had an impact on eGC condition, or whether it is associated with unfavorable outcomes for endothelial function in STEMI patients.

## 2. Materials and Methods

### 2.1. Study Population

In this study, 63 consecutive patients with a first onset of STEMI were recruited at the University of Luebeck in cooperation with the intensive care unit (ICU) of the Department of Cardiology and Angiology of the Sana-Kliniken-Luebeck hospital, Germany, in accordance with the Declaration of Helsinki and approved by the Local Ethics Committee (Case: 19-310). Informed consent was obtained from each patient. All patients received emergency coronary angiography and PCI as first-line therapy, with subsequent treatment in the ICU after STEMI was diagnosed. The choice of the preclinical emergency medication, as well as the diagnosis of STEMI, was based on the criteria of the Guidelines for the management of acute myocardial infarction in patients presenting with ST-segment elevation of the ESC [9]. Blood samples were collected during emergency PCI (hereafter termed the STEMI group). Sixty-three age- and sex-matched volunteers, without cardiovascular comorbidities, served as controls (hereafter termed the CTR group). Patients undergoing cardiopulmonary resuscitation or patients after gaining a return of spontaneous circulation (ROSC) were excluded, as were patients who died during or after PCI. Patients requiring extracorporeal membrane oxygenation/extracorporeal life support (ECMO/ECLS) were also excluded. Further exclusion criteria were aged below 18 years or pregnancy.

Serum samples from patients and controls were immediately treated according to the manufacturer’s (S-Monovette^®^, Sarstedt, Nümbrecht, Germany) instructions. Therefore, blood samples were kept at 4 °C and centrifuged at 2000× *g* for 10 min, within 60 min after collection. Afterwards the protruding serum was collected, snap-frozen, and stored at −80 °C.

For D2B analysis, patients were retrospectively divided into two groups: (i) D2B ≤ 60 min; (ii) D2B > 60 min. The D2B was defined as the time interval between the STEMI patient entering the emergency room and the time of the first balloon dilatation in the catheter laboratory. The cut-off of 60 min was chosen in accordance with the recommendations of the ESC guidelines for PCI-capable hospitals [8].

### 2.2. Cell Isolation and Culture

Primary human umbilical vein endothelial cells (HUVEC) were isolated (approved by the local ethical committee; Case: 18-325) and cultured as described previously [11,15]. Cells were cultured in HUVEC culture medium (Gibco Medium 199)  + 10% fetal calf serum (Gibco, Carlsbad, CA, USA)  +  penicillin/streptomycin 1% (Gibco, Carlsbad, CA, USA; 100 U/mL; 100 mg/mL)  +  heparin 5000 U/mL (Biochrom, Schaffhausen, Switzerland)  +  large vessel endothelial supplement 1% (Gibco, Carlsbad, CA, USA). Cell culture flasks were coated with 0.5% gelatin (Sigma-Aldrich, St. Louis, MO, USA) 1 h before seeding and cultivated at 37 °C, with 21% O_2_ and 5% CO_2_. For the experiments, HUVEC were cultivated on fibronectin-coated glass coverslips, to confluence for at least 4 days under standard cell culture conditions and were stimulated with 10% STEMI or CTR sera for 24 h prior to the experiment.

### 2.3. Atomic Force Microscopy

The height and stiffness of the eGC were determined by using the AFM nanoindentation technique, as described previously [11]. Indentation measurements were performed on living confluent HUVEC at 37 °C, using a Nanoscope Multimode-8 AFM (Bruker Nano GmbH, Berlin, Germany).

Briefly, a laser beam was aligned on the back of a gold-coated triangular cantilever (Novascan Technologies, Boone, NC, USA) with a mounted spherical tip (diameter 10 μm) and a nominal spring constant of 10 pN/nm. The cantilever indents the endothelial cell surface with a loading force of 0.5 nN. The reflection of a laser beam is used to quantify the cantilever deflection. The height of the eGC can be calculated by knowing the cantilever force, the piezo displacement, and the deflection sensitivity. For each patient and control serum, a total of 50–60 cells were measured. For each cell, 6–8 force distance curves (FDCs) were generated and averaged, resulting in *n* = 300 to 480 FDCs per individual. FDC data were collected with the Research NanoScope version 9.20 (64 bit; Bruker Nano GmbH). The stiffness and thickness of the eGC were calculated using the protein unfolding and nanoindentation analysis software Punias 3D (Version 1.0; Release 2.3; Copyright 2009).

### 2.4. Enzyme-Linked Immunosorbent Assay

In addition to the standard clinical laboratory examinations of the patient’s blood, additional parameters in the serum of the patients were determined via enzyme-linked immunosorbent assay (ELISA).

To determine the dissolved glycocalyx constituents, syndecan-1, heparan sulfate, and hyaluronic acid (hyaluronan) concentrations were measured (syndecan-1: Human CD138 ELISA kit, Diaclone Research, Besançon, France; catalog: 950.640.192/heparan sulfate: Human Heparan sulfate Proteoglycan (HSPG) ELISA Kit, MBS, San Diego, CA, USA; catalog: MBS2023323/hyaluronan: Hyaluronan Quantikine ELISA Kit, R&D Systems, Minneapolis, MN, USA; catalog: DHYAL0). Angiopoetin-2 was quantified using a Human Angiopoietin-2 Quantikine ELISA Kit (R&D Systems, Minneapolis, MN, USA; catalog: DANG20). Activation of the complement system was measured by quantifying the anaphylatoxins C3a and C5a (Thermo Fisher Scientific, Hamburg, Germany; C3a, catalog: BMS2089; C5a, catalog: BMS2088).

### 2.5. Nitric Oxide Product Measurements

For measuring total nitrate and nitrite in the STEMI and CTR sera, the chemiluminescence detector Sievers Nitric Oxide Analyzer (NOA-280i; GE Water & Process Technologies, Analytic Instruments; Boulder, CO, USA) was used. Subsequent procedures were performed according to the operation and maintenance manual (Firmware Version 3.00 and later) provided by the manufacturer. The assay is based on the reduction of all nitrates and nitrites into nitric oxide (NO) by vanadium (III) chloride. NO reacts with ozone inside the NOA-280i to produce nitrogen dioxide (NO_2_), which is sensitively detected by virtue of its chemiluminescence.

NO products (NOx) of the STEMI and CTR sera were analyzed by injecting 50 μL of each serum sample into a purge vessel containing a solution of vanadium (III) chloride (50 mmol/L; Sigma-Aldrich, Hamburg, Germany) in hydrochloric acid (HCl) (1 mol/L; Sigma-Aldrich, Hamburg, Germany) at 95 °C, continuously purged with a stream of nitrogen gas, connected to the NOA-280i. A gas bubbler between the purge vessel and the NOA-280i was filled with 15 mL of 1 M aqueous NaOH solution (Sigma-Aldrich, Hamburg, Germany) to prevent HCl vapors from entering the NOA-280i. Concentrations were calculated using the manufacturer’s NO Analysis Software for Liquid (Version 3.21/Liquid, GE Water & Process Technologies, Analytic Instruments; Boulder, CO, USA).

### 2.6. Statistical Analysis

Data were analyzed using IBM SPSS Statistics for Windows (IBM Corp., released 2020, Version 28.0.1 Armonk, New York, NY, USA) and the 2D graphic and biostatistics software GraphPad PRISM (Version 8.4.2, GraphPad Software Inc., San Diego, CA, USA). GraphPad PRISM was also used to prepare the figures. The Gaussian distribution was determined by a D’Agostino–Pearson omnibus normality test and presented graphically via quantile–quantile plot (Q–Q Plot). Data with no proven linearity were plotted, and the curves were fitted to determine the interrelationships of the function. The best-fit model was implemented. Data were tested for outliers before applying the statistical tests, using the ROUT outlier test based on the false discovery rate (FDR; Q value = 1%). Outliers were omitted from further analysis.

The differences between the two groups were analyzed using Student’s *t*-test for parametric values. For nonparametric values, the Mann–Whitney test (for unpaired data) or the Wilcoxon matched-pairs signed-rank test (for paired data) was applied. Group differences at the nominal scale level were measured using Cramer’s V. Categorical variables were compared using the chi-squared test. Correlations at the ordinal scale level were measured using Spearman and at the metric scale level using Pearson correlations (with Rho (r); coefficient of determination (R^2^)). With a sample size of *n* = 63, a statistical power of 0.8, and a significance level of α = 0.05, a correlation of r = 0.344 was needed for a significant result.

Patients were retrospectively divided into cohorts: (a) D2B ≤ 60 vs. >60 min, in accordance with the recommendations of the European Society of Cardiology guidelines (ESC) for PCI-capable hospitals [8]; (b) syndecan-1 levels ≤ 120 vs. >120 ng/mL, in accordance to the findings of Wernly et al. [13]. Killip classification was determined on admission and carried out according to Killip and Kimball (1967) [16].

## 3. Results

### 3.1. Characteristics of STEMI Group

The patients in the STEMI group were 64 (±13) years old on average and predominantly male (76%). All of them presented with at least one cardiovascular risk factor (CVRF: male sex, hypertension, diabetes, obesity, a positive family history of myocardial infarction, hyperlipidemia, uremia, or active smoking; see Table 1), and 65% had more than three CVRFs. The average D2B was 55.5 (±27.2) min (Table 1), whereby D2B was less than 60 min for 66% of patients and more than 90 min for 14%. In 69% of patients, a Killip classification of III or IV was calculated on admission, with 38% of the patients having a preserved left-ventricular ejection fraction (LVEF) of over 50%. Reduced LVEF was diagnosed in 16% of patients. The average systolic blood pressure was 85.4 (±14.7) mmHg with an average heart rate of 102 beats per second. Of the patients, 65% had an anterior infarction. Cardiac markers (troponin-t, creatine kinase (CK), lactate dehydrogenase (LDH), and pro-brain natriuretic peptide II (pro-BNP II)) as well as creatinine levels determined in laboratory examinations of the STEMI group, showed elevated levels overall compared to the laboratory reference values (Table 1). Markers for inflammatory processes such as C-reactive protein (CRP), leukocyte count, eGC components, and complement activation were increased in the STEMI group compared to healthy controls. On average, patients were hospitalized for 8 (±3) days before being discharged from the hospital.

### 3.2. STEMI Leads to eGC Damage and Endothelial Dysfunction

The nanomechanical properties (height and stiffness) of the eGC were quantified using the AFM nanoindentation technique. The height of the eGC was 38% lower in the STEMI group than in the CTR group (*p* < 0.001; Figure 1A). eGC stiffness was reduced by 17% in the STEMI group compared to CTR (*p* < 0.001; Figure 1B).

Overall, the levels of shed glycocalyx constituents measured via ELISA were elevated in the STEMI group; mean syndecan-1 levels in the CTR group were 35.5 ng/mL (±10.4 ng/mL), whereas the STEMI mean levels were about four times higher (136.72 ± 69.3 ng/mL; *p* < 0.0001; Figure 1C). Mean heparan sulfate levels were 4.6 ng/mL (±3.6 ng/mL) in the CTR group, with the amount of detected heparan sulfate being twice as high in the STEMI group (10.82 ± 8.6 ng/mL; *p* < 0.001; Figure 1D). Additionally, the measured levels of hyaluronic acid (hyaluronan) were elevated by 44% in the STEMI group compared to healthy CTR (*p* < 0.0001; Figure 1E). NOx were measured using the NOAnalyzer-280i. Levels of NOx were lower in the STEMI group than in the CTR group by about 34% (6.4 vs. 9.7; *p* < 0.001; Figure 1F). Correlations between the individual parameters were determined with the help of regression analyses. eGC height was positively associated with eGC stiffness (r = 0.918, *p* < 0.001; Figure 1G). Days until discharge from hospital were negatively correlated to both eGC height (r = 0.572; *p* < 0.001; Figure 1H) and eGC stiffness (r = 0.674; *p* < 0.001; Figure 1I). The eGC height of the STEMI group was lower in 97% of the cases, in direct comparison to the individual patient’s age- and sex-matched controls (Figure 1J). Notably, patient troponin levels were associated positively with hyaluronic acid concentrations (r = 0.285; *p* < 0.05)

### 3.3. D2B Time > 60 Min Leads to eGC Damage and Prolonged Hospitalization

Patients were divided into two cohorts according to their D2B time (≤60 vs. >60 min). In order to maintain a better overview, the group with a D2B time ≤ 60 min will be referred to as BELOW in the following, and the group with a D2B time > 60 min as the ABOVE group. Overall, 34% of the STEMI group had a D2B of >60 min (Table 2). There was no statistically significant difference between the two groups in terms of age, sex, or CVRF. There was a nonsignificant trend towards higher rates of Killip class III and IV (28.6% vs. 71.4%; *p* = 0.066) between the BELOW and the ABOVE group (Table 2). eGC height was reduced by 16% in the ABOVE compared to the BELOW group (124.9 vs. 106.8 nm; *p* < 0.001; Figure 2A). The stiffness of eGC was higher in the BELOW than in the ABOVE group (0.35 vs. 0.3 pN/nm; *p* < 0.001; Figure 2B).

Levels of syndecan-1 were elevated by 53% (*p* = 0.02; Figure 2C) and troponin levels were twice as high (*p* < 0.01; Figure 2D) in the ABOVE group. The NO concentration was higher in the BELOW group compared to ABOVE (6.14 vs. 4.67 mM; *p* < 0.01; Figure 2E). Puncture-to-balloon time, at 9.8 (±11.9) min, as well as door-to-needle time, at 25.1 (±21.6) min, were shorter in the BELOW than in the ABOVE group (Table 2). D2B negatively correlated with both eGC height (r = 0.516; *p* < 0.001; Figure 2F) and eGC stiffness (r = 0.586; *p* < 0.001; Figure 2G). A positive correlation was shown between the D2B and the number of days until discharge from hospital (r = 0.426; Figure 2H).

In addition to markers for vascular inflammation, such as eGC height and stiffness, further inflammatory markers, such as CRP and leukocyte count, showed elevated levels compared to the laboratory reference values (Table 2), although there were no significant differences between ABOVE and BELOW. A prolonged D2B, however, led to significant activation of the complement system in the ABOVE group, with elevated levels of C3a (972.2 ± 262.5 ng/mL, *p* < 0.001) and C5a (61.4 ± 22.7 ng/mL, *p* < 0.001) (Table 2).

### 3.4. High Syndecan-1 Levels Are Associated with Unfavorable Outcomes for eGC and Patients

Patients were divided into two cohorts according to their syndecan-1 levels, according to the findings of Wernly et al. (2019) [13]. In the following sections of this report, the group with syndecan-1 levels ≤ 120 ng/mL will be referred to as LOW and the group with syndecan-1 levels > 120 ng/mL as HIGH.

In the STEMI group, 54% had syndecan-1 levels >120 ng/mL (Table 3). Comparisons between the LOW and HIGH group showed no statistically significant differences in terms of age, sex, CVRF, Killip classification, or LVEF. There were, however, nonsignificant trends towards higher levels of ProBNP II (*p* = 0.069), creatinine (*p* = 0.067), and CRP (*p* = 0.08) in the HIGH compared to the LOW group (Table 3).

eGC height and stiffness were reduced in the HIGH compared to the LOW group: eGC height by 32% (151.3 vs. 104.4 nm; *p* < 0.001; Figure 3A) and eGC stiffness by 24% (0.38 vs. 0.29 pN/nm; *p* < 0.001; Figure 3B). In addition to the D2B time (63.9 ± 31.3 min; *p* = 0.006; Figure 3C), the puncture-to-balloon time (13.9 ± 9.7 min, *p* = 0.033; Table 3) of the HIGH group was longer than in the LOW group. Patients with syndecan-1 levels > 120 ng/mL stayed in the hospital longer than those in the LOW group (10 vs. 6 days until discharge from hospital; *p* < 0.001; Figure 3D). eGC height (r =0.924; *p* < 0.001; Figure 3E) and eGC stiffness (r = 0.811; *p* < 0.001; Figure 3F) correlated significantly with syndecan-1 levels. Furthermore, there was a linear relationship between the syndecan-1 levels and the days until discharge of r = 0.709 (*p* < 0.001; Figure 3G) as well as between syndecan-1 levels and the D2B (r = 0.637; Figure 3H). 

Levels of heparan sulfate (11.67 ± 2.5 ng/mL; *p* < 0.001; Table 3) and CK (759.1 ± 1767 U/l; *p* < 0.01; Table 3) were elevated in the HIGH group compared to the LOW group. There was a significant elevation of inflammatory markers in the HIGH group, compared to the laboratory reference values. Furthermore, the HIGH group showed an activation of the complement system, with elevated levels of C3a (963.6 ± 322.9 ng/mL, *p* < 0.001) and C5a (59.1 ± 29.5 ng/mL, *p* < 0.001) (Table 3).

## 4. Discussion

The current study aimed to investigate the relationship between D2B duration and eGC condition, with regard to endothelial function in the case of STEMI. Particular attention was paid to the nanomechanical properties (height/stiffness) of the eGC. Multiple studies have demonstrated the importance of the shortest possible period of time between the first medical contact and revascularization in acute myocardial infarction [8,17], but to our knowledge there are no published data showing a temporal link between myocardial infarction and endothelial injury. Using AFM to quantify the nanomechanical properties of the eGC, our data show that the endothelial surface is injured during STEMI. Furthermore, the deterioration of the eGC during STEMI leads to a reduced production of endothelium-derived relaxing factors, such as NO, indicating endothelial dysfunction. For the first time, we could demonstrate that a shorter D2B time is associated both with fewer changes in the nanomechanical properties of the eGC (r = 0.516) and lower concentrations of syndecan-1 (r = 0.637), indicating less damage to the eGC. The eGC damage and loss of endothelial function is significantly lower in the group of patients with a D2B of under 60 min. Furthermore, shorter D2B results in a shorter hospitalization. Although it has been shown that cardiac I/R severely damages the eGC, which can be detected by increased circulating levels of its principal constituents [13,18], until now there were no published data that have demonstrated a temporal relationship between eGC constituents and the D2B duration. A prolonged D2B may thus cause more severe damage to the eGC, which might be the “stumbling block” to severe cardiac IRI, ultimately leading to post-ischemic coronary microvascular dysfunction, cardiogenic shock, and death.

A reduction in eGC height and stiffness indicates eGC shedding [1,11]. Shedding of the eGC is known to be caused by different factors that are elevated and activated during cardiac IRI [19]. Those factors are associated with cardiac mechanical stress, generalized vascular trauma, and an increased inflammatory response [11,20]. Proinflammatory mediators, such as interleukins (IL) [21], catecholamines [22], angiopoetin-2 [23], CRP [24], leukocytes [20] including T-lymphocytes [25], and matrix metalloproteinases (MMP) [21], as well as tissue inhibitors of matrix metalloproteinases (TIMP) [26], or the complement system [27], are elevated during acute myocardial infarction. This inflammation-mediated response results in cell death of the ischemic tissue and subsequent long-term consequences, such as postinfarction heart failure with the hallmarks of cardiac fibrosis and heart dysfunction [28]. Progressive cardiac dysfunction is associated with CD4+ T-cell activation and transmigration, both leading to left ventricular remodeling and progressive cardiac dysfunction [25]. The activation of the immune system during a STEMI can be caused by several factors. On the one hand, it is most likely favored by pre-existing atherosclerosis. Wolf et al. explained that atherosclerosis, although traditionally regarded as a cholesterol storage disease, is additionally caused by a chronic low-grade inflammatory response that attracts cells of the innate and adaptive immune systems [29]. On the other hand, acute ischemia leads to a general upregulation of both innate and adaptive immunity, which can reinforce each other [30,31].

The extent and severity of atherosclerosis in coronary artery disease are closely related to elevated levels of MMP-1 and MMP-9, as well as TIMP-1 and IL-6 [32]. Increased blood levels of MMP-1, MMP-9, and IL-6 during acute myocardial infarction has already been demonstrated in several previous studies [26,32,33,34]. Since all of these factors have been shown to be directly related to atherosclerosis in coronary artery disease, it is reasonable to assume that they are also involved in the acute endothelial dysfunction shown in this study. A detailed analysis of these factors in relation to the development of eGC damage in the context of an extended D2B are unfortunately beyond the scope of this study but should be examined in future experimental setups to determine their individual influence on endothelial impairment.

However, some of the mentioned biomarkers activated in the context of STEMI are connected to eGC damage: interleukins (IL) [21], catecholamines [22], and CRP [24], as well as the complement system [27], were identified as factors influencing the eGC condition. The strong correlations between eGC impairment and the elevated levels of CRP, leukocyte count, and elevation of the complement anaphylatoxins indicate a proinflammatory response with CRP-induced MMP-9 expression [35]. Also, IL-6, which is known to be elevated during STEMI [34], has previously been shown to be associated with eGC damage and loss of eGC height [15]. The process of eGC shedding is further underpinned by elevated levels of the eGC components (syndecan-1, heparan sulfate, and hyaluronic acid) measured in the STEMI sera, indicating elevated levels of MMPs, and by the strong correlation between eGC height and stiffness (r = 0.918).

High syndecan-1 levels have been found to be an independent predictor for outcome in patients with STEMI, independent of the infarct-related myocardial injury [13] and are an independent predictor of mortality in cardiogenic shock [36,37]. Compared to healthy individuals, syndecan-1 concentrations are significantly higher in STEMI patients. This effect is mostly explained by the activation of MMPs, which have been shown to be commonly upregulated in cardiac IRI triggering glycocalyx damage [38]. In our study we examined the blood levels of eGC components. Here, these levels reflect the eGC impairment, not only of the hearts vasculature but of the whole body, as STEMI leads to a global whole body ischemia [2,37]. The proportion of eGC components that originally come from the cardiac vasculature is unfortunately almost impossible to comprehend. The effects shown in the present study are therefore to be understood as a change in the eGC condition of the entire vasculature and not only limited to the heart [2]. It should be noted, however, that the changes shown here occurred in the context of an acute ischemic event with consecutive activation of the immune system. Whether this also leads to long-term damage to the eGC or the impairment of endothelial function remains speculative.

In this context, it can be hypothesized that increased syndecan-1 indicates eGC shedding after STEMI, impairing eGC and vascular function and leading to adverse outcomes. The same applies to increased levels of other eGC components, such as heparan sulfate or hyaluronic acid. Syndecan-1 levels > 120 ng/mL have been shown to be independently associated with a higher 6-month mortality after STEMI [13]. In our cohort, 54% of the STEMI patients showed an elevation of this magnitude. Without analyzing mortality as an endpoint in this study, a prolonged hospital stay suggests that patients with an elevated syndecan-1 level > 120 ng/mL were significantly more severely ill than patients with lower syndecan-1 levels. Likewise, there was a strong interaction between the nanomechanical properties of the eGC and syndecan-1 levels, which further correlated with a prolonged D2B time, indicating a time dependency of the eGC damage during STEMI. This time dependency of eGC damage could be explained by an overall prolonged inflammatory response in the phase of chronic inflammation after myocardial ischemia [31].

The deterioration of the eGC also correlated with the degree of NO release—the hallmark for endothelial dysfunction [39]. In a functional endothelium, NO is released by the endothelial cells themselves and diffuses to adjacent vascular smooth muscle cells (VSMC), where it triggers vasodilation via cyclic guanosine monophosphate (cGMP)-dependent pathways [39]. Here, the reduction in NO production demonstrates the link between the altered nanomechanical properties of the eGC and the beginning of endothelial dysfunction during STEMI. Furthermore, it could be shown that NO production is decreased with an increasing D2B duration, which indicates additional damage to the endothelium over time during STEMI (see illustrative Figure 4).

The disruption to eGC integrity in cardiac IRI has been documented in the meantime [2], but, to date, no satisfactory cardioprotective therapy against IRI is available for daily clinical practice [40]. Here, protecting the eGC in the case of STEMI may result in less cardiac IRI.

Our study has established the basis for further investigations to illustrate the important role that the eGC plays in the development of endothelial dysfunction during cardiac IRI. Despite all previous knowledge, the eGC still represents an underestimated factor in the development of post-ischemic coronary microvascular dysfunction and cardiac IRI. On the one hand, this is due to the complex and multi-layered cell biological background and mechanisms of eGC damage in cardiac IRI [3], but, on the other, also to the limited translatability from basic research to clinical practice, both for diagnostic options and pharmacological approaches to managing IRI [40].

Pharmacological approaches were not addressed in the present study. Since all patients had a first-time event with previously untreated comorbidities, they can be described as drug-naïve. In the context of emergency treatment, all patients received the same standard medication, as suggested in the ESC guidelines [9]. Differences in dosage are not foreseen by the ESC, so balanced drug levels within the cohort were assumed. A treatment in the sense of a case–control model has not been established. Positive effects on the eGC in the context of acute myocardial infarction with the help of a recombinant syndecan-1 have been described before [11]. Answering the question of whether recombinant syndecan-1 or other therapeutic strategies, to prevent or repair damage of the eGC, have beneficial effects on endothelial function will be the subject of future studies.

Our AFM-based methodology is time consuming and sophisticated, which precludes analyzing considerably larger random sets of samples; however, there are approaches for meaningful analysis of the eGC status that can be probed in everyday clinical practice: For example, by visualizing the sublingual microcirculation, the integrity of the glycocalyx could be assessed indirectly [41] and could represent an important diagnostic tool to measure eGC integrity in the future, allowing further predictions of the outcome of STEMI patients [42]. The association between sublingual microcirculation parameters and eGC dimensions has already been demonstrated for critically ill patients [43]. It is now known that the study of microcirculation parameters and eGC dimensions is an important part of the assessment of septic patients [14]. So far, however, this need has not been demonstrated for patients during acute myocardial infarction. Here, we illustrate the time-dependent development of endothelial dysfunction in the event of STEMI and the important role the eGC condition plays in the endothelial function. Future clinical studies should evaluate the prognostic value of eGC protection or restoration, in the case of acute myocardial infarction.

## 5. Conclusions

Endothelial dysfunction, eGC shedding, and the D2B are associated, in a time-dependent manner after STEMI. In addition, levels of syndecan-1 and pro-inflammatory mediators correlate with prolonged D2B, eGC damage, and endothelial dysfunction and could therefore be important factors for the risk stratification of IRI. A combination of clinical evaluation of the eGC condition and levels of biomarkers, such as syndecan-1, might serve as important predictors for eGC impairment and the detection of endothelial dysfunction in STEMI patients. Future medical studies should investigate the impact of eGC damage on patient outcomes. In addition, strategies to protect or restore the eGC in acute cardiac ischemia should be tested, to improve patient care and prevent endothelial dysfunction during STEMI.

## Figures and Tables

**Figure 1 biomedicines-11-02924-f001:**
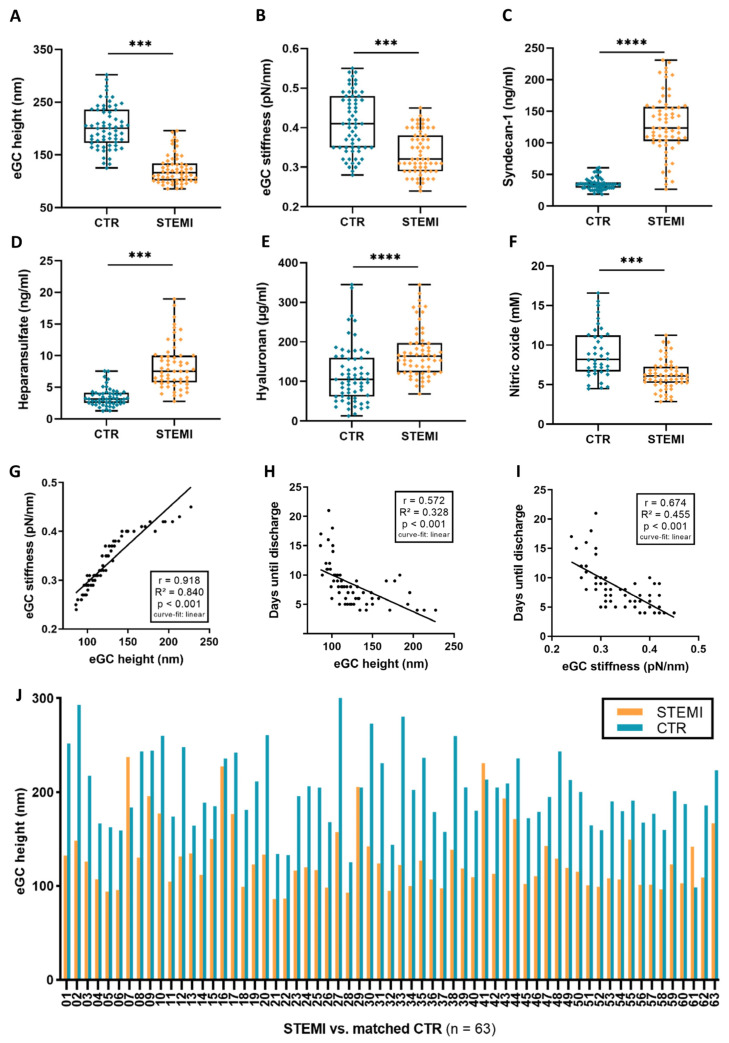
ST-elevation myocardial infarction (STEMI) leads to endothelial glycocalyx (eGC) damage and endothelial dysfunction. Endothelial glycocalyx (eGC) height (**A**) and eGC stiffness (**B**) measured via AFM nanoindentation technique. Each dot represents one patient/healthy control (*n* = 126; each dot shows a mean of 50–60 cells per patient). Serum levels of syndecan-1 (**C**), heparan sulfate (**D**) and hyaluronic acid (hyaluronan) (**E**) were measured via ELISA. Nitric oxide (NO) products (**F**) were quantified via NO-Analyzer-280i. A-F: data shown as mean ± SD. Correlation of eGC height vs. eGC stiffness (**G**) and days until discharge from hospital (**H**). Correlation of eGC stiffness vs. days until discharge from hospital (**I**). Direct comparison of individual eGC height of STEMI patients and age- and sex-matched controls (**J**). Groups: CTR (control) stimulation with cell culture media + 10% serum of healthy controls; STEMI (ST-elevation myocardial infarction) stimulation with media + 10% serum of STEMI patients. *p*-values: ****: *p* < 0.0001; ***: *p* < 0.001. Rho (r), *p*-values (*p*), coefficient of determination (R^2^), and curve-fit model shown for correlations.

**Figure 2 biomedicines-11-02924-f002:**
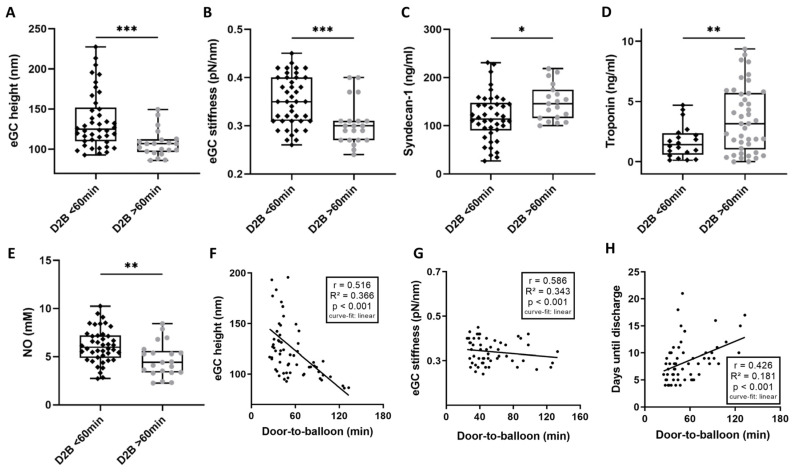
Door to balloon (D2B) time > 60 min leads to endothelial glycocalyx (eGC) damage and prolonged hospitalization. STEMI (ST-elevation myocardial infarction) patients were divided into cohorts: door-to-balloon time (D2B) ≤ 60 min and D2B > 60 min. Endothelial glycocalyx (eGC) height (**A**) and eGC stiffness (**B**) were measured via AFM nanoindentation technique. Each dot represents one patient/healthy control (n = 63; each dot shows a mean of 50–60 cells per patient). Quantification of syndecan-1 (**C**), troponin-t (**D**) and nitric oxide (**E**) serum levels. Correlation of D2B vs. eGC height (**F**), eGC stiffness (**G**) and days until discharge from hospital (**H**). *p*-values: *: *p* < 0.05; **: *p* < 0.01; ***: *p* < 0.001. Rho (r), *p*-values (*p*), coefficient of determination (R^2^) and curve-fit model shown for correlations.

**Figure 3 biomedicines-11-02924-f003:**
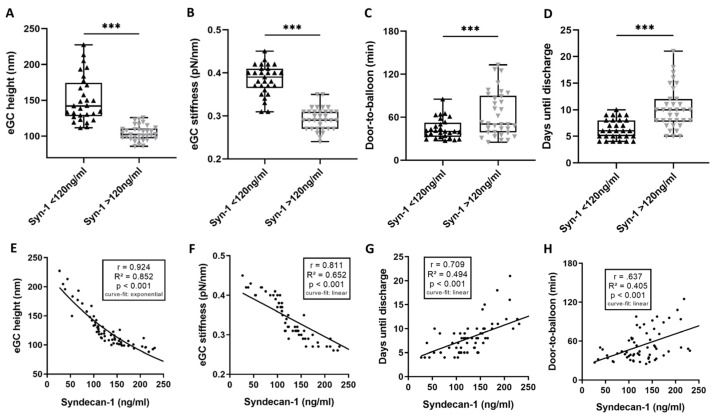
High syndecan-1 is associated with unfavorable outcomes for endothelial glycocalyx (eGC) and patients. STEMI (ST-elevation myocardial infarction) patients were divided into cohorts: (a) syndecan-1 levels ≤ 120 ng/mL and (b) syndecan-1 levels > 120 ng/mL. Endothelial glycocalyx (eGC) height (**A**) and eGC stiffness (**B**) were measured via AFM nanoindentation technique. Each dot represents one patient/healthy control (*n* = 63; each dot shows a mean of 50–60 cells per patient). Group differences between (a) vs. (b) concerning door-to-balloon (D2B) time (**C**) and days until discharge from hospital (**D**). Correlation of syndecan-1 levels vs. eGC height (**E**), eGC stiffness (**F**), days until discharge from hospital (**G**), and door-to-balloon time (**H**). *p*-values: ***: *p* < 0.001. Rho (r), *p*-values (*p*), coefficient of determination (R^2^) and curve-fit model shown for correlations.

**Figure 4 biomedicines-11-02924-f004:**
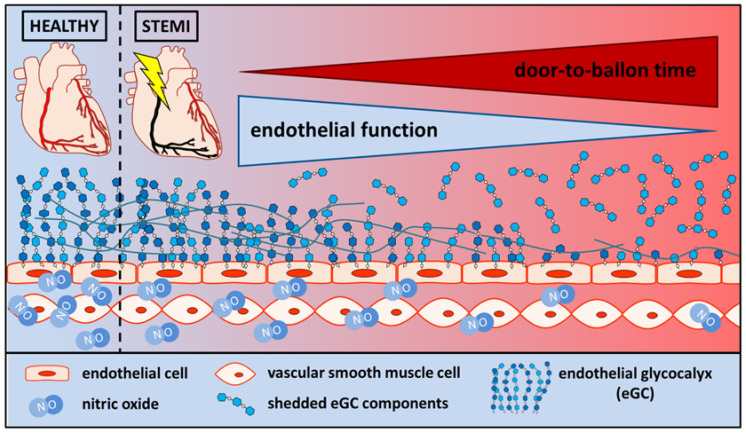
In healthy conditions, the vascular endothelial function is preserved, with adequate secretion of the endothelium-derived relaxing factor nitric oxide. Following ST-elevation myocardial infarction, endothelial function is impaired, with limited nitric oxide production. Endothelial function decreases as the door-to-balloon time increases. In addition, the eGC components are shed and become measurable in the blood stream.

**Table 1 biomedicines-11-02924-t001:** Characteristics of STEMI group.

Parameter		STEMI Group (n = 63)
Age (years)		64 (13)
Hypertension (%)		79.0
Diabetes (%)		65.5
BMI (kg/m^2^)		27.4 (4.5)
Pos. family history (%)		86.7
Hyperlipidemia (%)		88.9
Smoking (%)		62.3
Door-to-balloon time (min)		55.5 (27.2)
Puncture-to-balloon (min)		11.7 (7.8)
Door-to-needle (min)		31.3 (16.3)
Killip class (%)	I–II	30.2
	III–IV	69.9
LV EF (goups)	<40%	15.6
	40–50%	44.4
	>50%	37.8
Days until discharge		8.4 (3.7)
Troponin max. (pg/mL)	[Ref.: 0–14]	3053.0 (2653)
CK max. (U/L)	[Ref.: 20–200]	1100.8 (1464)
LDH max. (U/L)	[Ref.: 135–225]	446.6 (318.9)
Pro BNP II (pg/mL)	[Ref.: 0–121]	1972.7 (2614)
Creatinine (mg/dL)	[Ref.: 0.7–1.2]	2.19 (0.9)
Inflammatory markers		
CRP max. (mg/L)	[Ref.: 0–5]	45.4 (48.0)
Leukocytes (Gpt/L)	[Ref.: 4–9]	17.41 (4.6)
eGC height (nm)		125.9 (32.8)
eGC stiffness (pN/nm)		0.34 (0.05)
Syndecan-1 (ng/mL)		136.72 (69.3)
Heparan sulfate (ng/mL)		10.82 (8.6)
Hyaluronic acid (µg/mL)		182.9 (85.9)
Nitric oxide (mM)		6.39 (2.3)
Angiopoetin-2 (ng/mL)		18.5 (8.7)
C3a (ng/mL)		676.0 (343.4)
C5a (ng/mL)		36.2 (27.0)

STEMI: ST-elevation myocardial infarction; BMI: body mass index; Pos. family history: positive family history of myocardial infarction; LV EF: left ventricular ejection fraction; CK: creatine kinase; LDH: lactate dehydrogenase; BNP: brain natriuretic peptide; CRP: c-reactive protein; C3a: complement factor C3a; C5a: complement factor C5a; eGC: endothelial glycocalyx. Data shown as mean ± standard deviation (m ± SD); categorical data as percentage (%). Ref.: laboratory reference values.

**Table 2 biomedicines-11-02924-t002:** Characteristics of patients with door-to-balloon time ≤ 60 min vs. >60 min.

Parameter		D2B ≤ 60 min (*n* = 42)	D2B > 60 min (*n* = 21)	*p*-Value
Age (years)		66 (11)	63 (12)	0.422
Door-to-balloon time (min)		39.2 (21.7)	88.0 (8.4)	**<0.001**
Puncture-to-balloon (min)		9.8 (11.9)	17.0 (4.6)	**<0.01**
Door-to-needle (min)		25.1 (21.6)	43.7 (7.4)	**<0.001**
Killip class	I–II	31.0	28.6	0.066
	III–IV	69.0	71.4	
Days until discharge		7.9 (3.4)	9.3 (3.8)	0.162
Troponin max. (ng/mL)		1.7 (2.2)	3.5 (2.7)	**<0.01**
CK max. (U/L)		800.5 (1350)	1250.9 (1511)	0.253
LDH max. (U/L)		413.9 (275.4)	462.5 (340.1)	0.580
Pro BNP II (pg/mL)		1752.7 (2993)	2440.3 (1611)	0.551
Creatinine (mg/dL)		2.08 (1.4)	2.39 (0.5)	0.237
Inflammatory markers				
CRP max. (mg/L)		40.9 (58.3)	54.4 (42.1)	0.296
Leukocytes (Gpt/L)		11.54 (4.0)	12.37 (5.7)	0.634
eGC height (nm)		124.9 (16.9)	106.8 (34.9)	**<0.001**
eGC stiffness (pN/nm)		0.35 (0.04)	0.30 (0.05)	**<0.001**
Syndecan-1 (ng/mL)		118.11 (88.1)	173.95 (49.0)	**0.02**
Heparan sulfate (ng/mL)		7.99 (3.2)	12.24 (10.1)	0.065
Hyaluronic acid (µg/mL)		174.9 (90.8)	198.9 (83.3)	0.300
Nitric oxide (mM)		6.12 (3.6)	4.88 (2.1)	**<0.01**
Angiopoetin-2 (ng/mL)		17.7 (9.6)	20.1 (8.3)	0.302
C3a (ng/mL)		643.0 (385.3)	972.2 (262.5)	**<0.001**
C5a (ng/mL)		38.3 (28.7)	61.4 (22.7)	<0.001

D2B: door-to-balloon time; CK: creatine kinase; LDH: lactate dehydrogenase; BNP: brain natriuretic peptide; CRP: c-reactive protein; C3a: complement factor C3a; C5a: complement factor C5a; eGC: endothelial glycocalyx. Data shown as mean ± standard deviation (m ± SD); categorical data as percentage (%). *p*-values are shown in bold for variables with *p* < 0.05.

**Table 3 biomedicines-11-02924-t003:** Characteristics of patients with syndecan-1 levels ≤120 ng/mL vs. >120 ng/mL.

Parameter		Syn-1 ≤ 120 ng/mL (*n* = 29)	Syn-1 > 120 ng/mL (*n* = 34)	*p*-Value
Age (years)		66 (13)	63 (12)	0.306
Door-to-balloon time (min)		45.5 (17.1)	63.9 (31.3)	**<0.01**
Puncture-to-balloon (min)		9.5 (4.7)	13.9 (9.7)	**0.033**
Door-to-needle (min)		30.2 (11.5)	32.3 (19.6)	0.627
Killip class	I–II	31.0	29.4	0.290
	III–IV	69.0	70.6	
Days until discharge		6.3	10.2	**<0.001**
Troponin max. (ng/mL)		2.8 (3.0)	3.3 (2.4)	0.467
CK max. (U/L)		239.0 (861.9)	759.1 (1767)	**<0.01**
LDH max. (U/L)		386.4 (265.1)	497.7 (354.3)	0.176
Pro BNP II (pg/mL)		986.8 (1279)	2882.9 (3210)	0.069
Creatinine (mg/dL)		2.43 (1.4)	1.98 (0.2)	0.067
Inflammatory markers				
CRP max. (mg/L)		43.8 (39.7)	46.8 (54.7)	0.080
Leukocytes (Gpt/L)		11.16 (3.7)	12.24 (5.2)	0.357
eGC height (nm)		151.3 (32.0)	104.4 (10.5)	**<0.001**
eGC stiffness (pN/nm)		0.38 (0.03)	0.29 (0.02)	**<0.001**
Syndecan-1 (ng/mL)		88.93 (28.3)	177.49 (67.9)	**<0.001**
Heparan sulfate (ng/mL)		6.88 (10.8)	11.67 (2.5)	**<0.001**
Hyaluronic acid (µg/mL)		198.4 (107.2)	169.7 (60.8)	0.188
Nitric oxide (mM)		6.14 (2.2)	6.60 (2.3)	0.468
Angiopoetin-2 (ng/mL)		20.1 (9.1)	17.2 (8.4)	0.183
C3a (ng/mL)		505.5 (144.6)	963.6 (322.9)	<0.001
C5a (ng/mL)		30.5 (11.6)	59.1 (29.5)	**<0.001**

Syn-1: syndecan-1; CK: creatine kinase; LDH: lactate dehydrogenase; BNP: brain natriuretic peptide; CRP: c-reactive protein; C3a: complement factor C3a; C5a: complement factor C5a; eGC: endothelial glycocalyx. Data shown as mean ± standard deviation (m ± SD); categorical data as percentage (%). *p*-values are shown in bold for variables with *p* < 0.05.

## Data Availability

The datasets used and/or analyzed during the current study are available from the corresponding author on reasonable request.

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
