# Peer review of "Prolonged Door-to-Balloon Time Leads to Endothelial Glycocalyx Damage and Endothelial Dysfunction in Patients with ST-Elevation Myocardial Infarction"

_biomedicines, 2023, doi:10.3390/biomedicines11112924_

Round 1
Reviewer 1 Report
Comments and Suggestions for Authors
The manuscript entitled, “Prolonged Door-to-Balloon time leads to endothelial glycocalyx damage and endothelial dysfunction in patients with ST-Elevation Myocardial Infarction" by Vahldieck et al., discussed the relationship between D2B-duration and endothelial function during STEMI. A thorough investigation was carried out on the 126 individuals in this study. Following the application of patient/control sera to endothelial cells, the nanomechanical properties of the endothelial glycocalyx (eGC) were examined. The components of eGC were assessed using the ELISA, whereas the levels of nitric oxide (NO) were measured using a chemiluminescence-based method. Both eGC height and stiffness, as well as NO-concentration, were significantly reduced after STEMI (both, p<0.001). Authors described that the augmentation of D2B resulted in a pronounced shedding of eGC accompanied by impaired endothelial function within a temporal framework. The components of eGC and pro-inflammatory mediators exhibit a correlation with prolonged door-to-balloon time (D2B), suggesting the presence of a time-dependent immune response in patients with ST-segment elevation myocardial infarction (STEMI) accompanied by reduced nitric oxide (NO) concentration. They concluded that D2B plays a pivotal role in the development of endothelial glycocalyx injury in patients with ST-segment elevation myocardial infarction (STEMI).
While I agree that this is a necessary and appropriate manuscript, I have some suggestions for improving the article so that it provides a more thorough analysis of the topic.
Comments:
1 The English of manuscript can be polished (minor).
2, At least one illustrative figure may be provided as to highlight the summary of this study.
3, In conclusion, write importance and future prospective of this review.
4, Role if immune cells are also very important factor in Myocardial Infarction, therefore I would suggest adding few citations to put comprehensive view of this topic (PMID: 36093172; PMID: 27031276; PMID: 34389841; PMID: 36337927; PMID: 30653442 etc.)
5, Figure 1,2,3 quality may be improved with high resolution images (minor).
Comments on the Quality of English LanguageMinor editing of English language required
Author Response
Reviewer 1 comments:
- The English of manuscript can be polished (minor).
Response:
The manuscript has already undergone professional language editing
- At least one illustrative figure may be provided as to highlight the summary of this study.
Response:
Thank you for this idea. We have now included a cartoon (Fig 4.) to highlight the summary of this study
- In conclusion, write importance and future prospective of this review.
Response:
The appropriate changes have now been made in the revised manuscript.
Page 15, lines 519-522:
Future medical studies should investigate the impact of eGC damage on patient outcomes. In addition, strategies to protect or restore the eGC in acute cardiac ischemia should be tested to improve patient care and prevent endothelial dysfunction during STEMI.
- Role if immune cells are also very important factor in Myocardial Infarction, therefore I would suggest adding few citations to put comprehensive view of this topic (PMID: 36093172; PMID: 27031276; PMID: 34389841; PMID: 36337927; PMID: 30653442 etc.)
Response:
We thank the reviewer for suggesting these helpful citations - appropriate amendments have been made in the revised manuscript to further highlight the important role of immune cells. We have added the suggested papers to the discussion.
Page 12 lines 394-409:
Proinflammatory mediators such as interleukins (IL) [21], catecholamines [22], angiopoetin-2 [23], CRP [24], leukocytes [20] including T-lymphocytres [25], matrix metalloproteinases (MMP) [21] as well as Tissue Inhibitors of Matrix Metalloproteinases (TIMP) [26], or the complement system [27] are elevated during acute myocardial infarction. This inflammation-mediated response results in cell death of the ischemic tissue and subsequent long-term consequences such as postinfarction heart failure with the hallmarks of cardiac fibrosis and heart dysfunction [28]. Progressive cardiac dysfunction is associated with a CD4+ T-cell activation and transmigration both leading to left-ventricular remodeling and progressive cardiac dysfunction [25]. Activation of the immune system during a STEMI can be caused by several factors: On the one hand, it is most likely favored by pre-existing atherosclerosis. Wolf et al. explained that atherosclerosis, although traditionally regarded as a cholesterol storage disease, is additionally caused by a by a chronic, low-grade inflammatory response that attracts cells of the innate and adaptive immune systems [29]. On the other hand, acute ischemia leads to a general upregulation of both innate and adaptive immunity, which can reinforce each other [30, 31].
- Figure 1,2,3 quality may be improved with high resolution images (minor).
Response:
Thank you for this helpful advice. We will submit the figures in a better resolution.
Reviewer 2 Report
Comments and Suggestions for Authors
The article begins by highlighting the importance of the eGC and the cellular cortex (CTX) in providing a protective barrier and responsive hub for endothelial cells. The eGC and CTX are dynamic structures that can adapt their mechanical properties to changes in the bloodstream. The article suggests that damage to these structures, such as through ischemia-reperfusion injury (IRI), can lead to endothelial dysfunction.
The authors mention that previous studies have demonstrated isolated eGC damage in the setting of acute myocardial infarction, but the functionality of the endothelium itself has not been considered. They also mention that rapid revascularization has been shown to improve outcomes in ST-elevation myocardial infarction (STEMI), and the preferred mode of revascularization is primary percutaneous coronary intervention (PCI).
The article discusses the importance of door-to-balloon time (D2B), which refers to the time interval between a patient entering the medical system and revascularization to open the occluded vessel. A shorter D2B is associated with better prognosis. The authors mention that despite successful restoration of coronary blood flow in PCI procedures, perfusion to the distal coronary microvasculature is not always fully restored.
The authors introduce syndecan-1 as a biomarker for eGC condition. The authors suggest that biomarkers for endothelial damage, such as syndecan-1, have not yet been widely used in the context of acute cardiac ischemia.
In the Materials and Methods section, the reviewer recommends adding the speed at which the samples were centrifuged in lines 106-107. Furthermore, in order to confidently discuss endothelial dysfunction, it is necessary to prove at least its pro-inflammatory activation by measuring levels of pro-inflammatory cytokines interleukin 6 and 8, as well as levels of anti-angiogenic molecules PAI-1, TIMP1, and TIMP2 (as the main ones). It is desirable to demonstrate the presence of endothelial-mesenchymal transition and disruption of endothelial mechanotransduction. Perhaps the authors will plan some of the proposed experiments for future studies, while implementing others in this article.
Overall, the article provides a clear introduction to the topic and highlights the importance of the eGC, D2B duration, and endothelial dysfunction in acute cardiac ischemia.
Author Response
Reviewer 2 comments:
- In the Materials and Methods section, the reviewer recommends adding the speed at which the samples were centrifuged in lines 106-107.
Response:
Thank you for this advice. The missing information are now included in the methods section of the revised manuscript.
Page 3 lines 105-109:
Serum samples from patients and controls were immediately treated according to the manufacturer’s (S-Monovette®, Sarstedt, Nümbrecht, Germany) instructions: Therefore, blood samples were kept at 4°C and centrifuged at 2000 * g for 10 minutes within 60 min after collection. Afterwards the protruding serum was collected, snap-frozen and stored at -80°C.
- Furthermore, in order to confidently discuss endothelial dysfunction, it is necessary to prove at least its pro-inflammatory activation by measuring levels of pro-inflammatory cytokines interleukin 6 and 8, as well as levels of anti-angiogenic molecules PAI-1, TIMP1, and TIMP2 (as the main ones).
Response:
We fully agree with the reviewer that pro-inflammatory cytokines are major players in the context endothelial dysfunction. In the present study we could only concentrate on CRP, Leukocytes, eGC components (Syndecan-1, Heparan sulfate, Hyaluronic acid) and complement anaphylatoxins (C3a and C5a) which will be extended in future studies on this topic. However, in the revised version of the manuscript we have now included a detailed discussion on pro-inflammatory factors:
Page 12, lines 394-398:
Proinflammatory mediators such as interleukins (IL) [21], catecholamines [22], angiopoetin-2 [23], CRP [24], leukocytes [20] including T-lymphocytres [25], matrix metalloproteinases (MMP) [21] as well as Tissue Inhibitors of Matrix Metalloproteinases (TIMP) [26], or the complement system [27] are elevated during acute myocardial infarction.
Page 12 lines 410-427:
The extent and severity of atherosclerosis in coronary artery disease are closely related to elevated levels of MMP-1, MMP-9, as well as TIMP-1 and IL-6 [32]. Increased blood levels of MMP-1, MMP-9, IL-6 during acute myocardial infarction has already been demonstrated in several previous studies [26, 32–34]. Since all of these factors have been shown to be directly related to atherosclerosis in coronary artery disease, it is reasonable to assume that they are also involved in the acute endothelial dysfunction shown in this study. A detailed analysis of these factors in relation to the development of eGC damage in the context of an extended D2B are unfortunately beyond the scope of this study, but should be examined in future experimental setups to determine their individual influence on endothelial impairment.
However, some of the mentioned biomarkers activated in the context of STEMI are connected to eGC damage: Interleukins (IL) [21], catecholamines [22], CRP [24], as well as the complement system [27] were identified as factors influencing the eGC condition. The strong correlations between eGC impairment and the elevated levels of CRP, leukocyte count, and elevation of the complement anaphylatoxins indicate a proinflammatory response with CRP induced MMP-9 expression [35]. Also IL-6 which is known to be elevated during STEMI [34] has before been shown to be associated with eGC damage and loss of eGC height [15].
- It is desirable to demonstrate the presence of endothelial-mesenchymal transition and disruption of endothelial mechanotransduction. Perhaps the authors will plan some of the proposed experiments for future studies, while implementing others in this article.
Response:
Thank you for this good suggestion. We agree that the endothelial-mesenchymal transition and disruption of endothelial mechanotransduction is an important issue. It would be existing to include these topics in future projects. We hope for your understanding that an implementation in the present article would require a series of extensive experiment that would extend the time frame and scope of the present study. We will include your idea with great gratitude in our further experimental planning.
Reviewer 3 Report
Comments and Suggestions for Authors
This is a study investigating about the relationship between MI and endothelial glycocalyx damage and endothelial dysfunction. Authors demonstrated that prolonged door-to-balloon time was associated with endothelial glycocalyx damage and endothelial dysfunction. The findings were intriguing. There were several issues to be addressed.
# Age is the most impactful factor for endothelial function. The information about subject age in each group should be added (also control).
# Which endothelium is the parameter for endothelial glycocalyx and endothelial dysfunction derived from? Indeed, the difference in door-to-balloon time might affect only the limited area in the heart.
# Were there any associations between endothelial glycocalyx or endothelial dysfunction and myocardial injury parameters such as troponin?
# How about the effect of some drugs?
Comments on the Quality of English LanguageNo comment.
Author Response
Reviewer 3 comments:
- Age is the most impactful factor for endothelial function. The information about subject age in each group should be added (also control).
Response:
We thank the reviewer for this point. We have now added information about the age of the respective groups in the tables. The mean age of the control group is identical to the STEMI group, as the controls were age- and sex-matched.
Page 6, Table 1
Page 8, Table 2
Page 10, Table 3
- 2. Which endothelium is the parameter for endothelial glycocalyx and endothelial dysfunction derived from? Indeed, the difference in door-to-balloon time might affect only the limited area in the heart.
Response:
This is indeed one of the key questions in this context and worth to discuss.
To be honest – it is not known at the moment. But from many studies and experiments we can extrapolate that the general activation of the immune system leads to general activation of the endothelium with the consequence of endothelial dysfunction. We partly addressed this complex issue in the discussion.
Page 13 lines 436-445:
In our study we examined the blood levels of eGC components. Here, these levels reflect the eGC impairment not only of the hearts vasculature but of the whole body as STEMI leads to a global whole body ischemia [2], [37]. The proportion of eGC components that originally come from the cardiac vasculature is unfortunately almost impossible to comprehend. The effects shown in the present study are therefore to be understood as a change in the eGC condition of the entire vasculature and not only limited to the heart [2]. It should be noted, however, that the changes shown here occurred in the context of an acute ischemic event with consecutive activation of the immune system. Whether this also leads to long-term damage to the eGC or impairment of endothelial function remains speculative.
- Were there any associations between endothelial glycocalyx or endothelial dysfunction and myocardial injury parameters such as troponin?
Response:
Yes, there was indeed an association between the cardiac troponin and hyaluronic acid being one of the analyzed eGC components in this study. We added this information in the results of the revised manuscript.
Page 5, lines 245-246:
Of note, patient troponin levels associated positively with hyaluronic acid concentrations (r = 0.285; p < 0.05)
- How about the effect of some drugs?
Response:
We agree that drugs are an important issue. and the potential influence of pharmaceuticals can be very large. However, in the present study, the influence of individual medication could not be taken into account for the following reasons: the STEMI group in our work had not received any medication before the acute event, and all patients received the same treatment during the STEMI.
Thus, we assume similar effects of the drugs in the individual patients. A classic distribution into a treatment group and a control group could not be made. We have addressed this and further details in the discussion.
Page 14. Lines 487-496:
Pharmacological approaches were not addressed in the present study. Since all patients had a first-time event with previously untreated comorbidities, they can be described as drug-naïve. In the context of emergency treatment, all patients received the same standard medication as suggested in the ESC guideline [9]. Differences in dosage are not foreseen by the ESC, so balanced drug levels within the cohort were assumed. A treatment in the sense of a case-control model has not been established. Positive effects on eGC in the context of acute myocardial infarction with the help of a recombinant syndecan-1 been described before [11]. Answering the question of whether recombinant syndecan-1 or other therapeutic strategies to prevent or repair damage of the eGC have beneficial effects on endothelial function will be subject of future studies.
Round 2
Reviewer 3 Report
Comments and Suggestions for Authors
The revised manuscript was corrected appropriately.
Comments on the Quality of English LanguageNo comment